# Childhood factors associated with suicidal ideation among South African youth: A 28-year longitudinal study of the Birth to Twenty Plus cohort

Massimiliano Orri [1,2]*, Marilyn N. Ahun [3,4], Sara Naicker [5], Sahba Besharati [6,7], Linda M. Richter [5]

1 McGill Group for Suicide Studies, Douglas Mental Health University Institute, Department of Psychiatry, McGill University, Montréal, Canada, 2 Bordeaux Population Health Research Centre, Inserm U1219, University of Bordeaux, Bordeaux, France, 3 Department of Social and Preventive Medicine, Université de Montréal School of Public Health, Montréal, Canada, 4 Department of Global Health and Population, Harvard T.H. Chan School of Public Health, Boston, Massachusetts, United States of America, 5 DSI-NRF Centre of Excellence in Human Development, University of the Witwatersrand, Johannesburg, South Africa, 6 Department of Psychology, School of Human and Community Development, University of the Witwatersrand, Johannesburg, South Africa, 7 CIFAR Azrieli Global Scholars Program, CIFAR, Toronto, Canada

* massimiliano.orri@mcgill.ca

## Abstract

### Background

Although early life factors are associated with increased suicide risk in youth, there is a dearth of research on these associations for individuals growing up in disadvantaged socio-economic contexts, particularly in low- and middle-income countries (LMICs). We documented the association between individual, familial, and environmental factors in childhood with suicidal ideation among South African youth.

### Methods and findings

We used data from 2,020 participants in the Birth to Twenty Plus (Bt20+) study, a South African cohort following children born in Soweto, Johannesburg from birth (1990) to age 28 years (2018). Suicidal ideation was self-reported at ages 14, 17, 22, and 28 years, and the primary outcome of interest was suicidal ideation reported at any age. We assessed individual, familial, and socioeconomic characteristics at childbirth and during infancy, adverse childhood experiences (ACEs) between ages 5 and 13 years, and externalizing and internalizing problems between 5 and 10 years. We estimated odds ratios (ORs) of suicidal ideation for individuals exposed to selected childhood factors using logistic regression. Lifetime suicidal ideation was reported by 469 (23.2%) participants, with a 1.7:1 female/male ratio. Suicidal ideation rates peaked at age 17 and decreased thereafter. Socioeconomic adversity, low birth weight, higher birth order (i.e., increase in the order of birth in the family: first, second, third, fourth, or later born child), ACEs, and childhood externalizing problems were associated with suicidal ideation, differently patterned among males and females.

**Data Availability Statement:** Bt20+ is housed in the DSI-NRF Centre of Excellence in Human Development at the University of the Witwatersrand and requests for data can be made

through https://www.wits.ac.za/coe-human/open-access-datasets/.

**Funding:** This study is funded by a grant from the EU's Horizon 2020 research and innovation programme under the Marie Skłodowska-Curie grant agreement no. 793396 (MO). The 28-year data collection was funded by the Bill & Melinda Gates Foundation OPP 1164115 (LMR) The funders had no role in study design, data collection and analysis, decision to publish, or preparation of the manuscript.

**Competing interests:** The authors have declared that no competing interests exist.

**Abbreviations:** ACE, adverse childhood experience; Bt20+, Birth to Twenty Plus; HIC, high-income country; LMIC, low- and middle-income country; OR, odds ratio; RR, risk ratio; SACAS, South African Child Assessment Schedule; SD, standard deviation; STROBE, Strengthening the Reporting of Observational Studies in Epidemiology.

Socioeconomic adversity (OR 1.13, CI 1.01 to 1.27, $P$ = 0.031) was significantly associated with suicidal ideation among males only, while birth weight (OR 1.20, CI 1.02 to 1.41, $P$ = 0.03), ACEs (OR 1.11, CI 1.01 to 1.21, $P$ = 0.030), and higher birth order (OR 1.15, CI 1.07 to 1.243, $P$ < 0.001) were significantly associated with suicidal ideation among females only. Externalizing problems in childhood were significantly associated with suicidal ideation among both males (OR 1.23, 1.08 to 1.40, $P$ = 0.002) and females (OR 1.16, CI 1.03 to 1.30, $P$ = 0.011). Main limitations of the study are the high attrition rate (62% of the original sample was included in this analysis) and the heterogeneity in the measurements of suicidal ideation.

## Conclusions

In this study from South Africa, we observed that early life social and environmental adversities as well as childhood externalizing problems are associated with increased risk of suicidal ideation during adolescence and early adulthood.

## Author summary

### Why was this study done?

- Identifying childhood risk factors for suicidal ideation is key to implement population-based strategies to prevent suicide starting early in life.

- The literature on suicide-related outcomes in low- and middle-income countries (LMICs) is limited, with only 1 prior population-based longitudinal study from Brazil.

- To the best of our knowledge, no longitudinal study has been conducted in LMICs on the African continent, with evidence on risk factors for suicide-related outcomes almost uniquely relying on small cross-sectional studies and lacking information on childhood predictors.

### What did the researchers do and find?

- The authors conducted secondary analysis on data from a longitudinal population-based cohort—the Birth to Twenty Plus (Bt20+) cohort—which is the largest and longest running birth cohort in sub-Saharan Africa.

- Among the 2,020 participants followed up from birth (1990) to age 28 years (2018), 469 (23.2%) reported suicidal ideation between ages 14 and 28 years, with a peak in prevalence at age 17 years and an overall 1.7:1 female/male ratio.

- Socioeconomic adversity at the time of birth was associated with suicidal ideation among males only, while low birth weight, adverse childhood experiences (ACEs), and higher birth order were associated with suicidal ideation among females only. Externalizing problems in childhood were associated with suicidal ideation among both males and females.

## What do these findings mean?

- Addressing widespread social and environmental adversities as well as childhood externalizing problems at the population level could potentially be of interest in suicide prevention efforts in South Africa and similar LMIC contexts.

- Considering sex differences may be important to optimize prevention efforts.

- Due to attrition, this study was conducted on 62% of the initial representative sample. This may influence the generalizability of the findings to the initial population.

## Introduction

Suicide is an important cause of mortality worldwide, accounting for 800,000 deaths each year [1]. Among youth aged 15 to 25 years, suicide ranks as the second or third most common cause of death in most countries [2]. Suicidal ideation—the consideration of or desire to end one's own life [2]—is a strong predictor of subsequent suicidal acts [3]. In the United States, 1 in 5 youth experiencing suicidal ideation transition to elaborate concrete suicidal plans within a year, and 60% of those having such plans attempt suicide in this time frame [3]. These data suggest that preventing suicidal ideation is critical to reduce suicide risk. The burden of youth suicide is disproportionally high in low- and middle-income countries (LMICs), where nearly 90% of the world's youth live [4] and 78% of all suicides occur [5]. In particular, young people in LMICs in the African region have the highest prevalence of suicidal ideation, with 1 in 5 adolescents reporting seriously considering suicide in the past 12 months [6]. Evidence suggests that these high rates are likely to be underestimated [7]. Despite this elevated prevalence, research on suicide prevention in LMICs accounts for only a small fraction of the available evidence [8]. This strongly limits our understanding of risk factors for suicide-related outcomes (i.e., suicidal ideation, suicide attempt, and death by suicide) and precludes the development of tailored public health suicide prevention strategies in LMIC contexts.

Beyond the contribution of childhood externalizing problems (i.e., behaviors such as conduct disorder and hyperactivity/impulsivity that are overt and can result in conflict with others) [9] and internalizing problems (i.e., emotional symptoms such as anxiety or depression which reflect internal distress) [9–15], longitudinal studies conducted in high-income countries (HICs) have shown that early life factors—including perinatal and childhood factors—play an important role in increasing vulnerability to suicidal ideation and suicide attempts during the life course [15,16]. For example, a recent meta-analysis found that exposure to socioeconomic adversity (indexed as low socioeconomic status, low maternal age, or low parental education at childbirth) is associated with increased suicide risk later in life [16]. Similarly, early growth deficits as indexed by low birth weight were associated with increased suicide risk in prior HIC studies [16,17]. Adverse childhood experiences (ACEs), such as exposure to violence and abuse/neglect [18–20], family difficulties (e.g., early separation and single-parent families) [18], and parental substance use problems [21], have all consistently been associated with mental health problems and suicide risk in HIC contexts [22].

However, it is unknown if and how such childhood factors are associated with suicide risk in LMIC contexts, which are markedly different from HIC contexts in a number of ways. First, the population prevalence of most childhood risk factors—including socioeconomic adversity and low birth weight—in LMICs is much higher than in HICs [23]. Second, several studies

have shown that exposure to ACEs is more common in LMICs than in HICs. For example, in South Africa and Brazil, more than 85% of children reported at least 1 ACE [24,25], compared to 46% of children in the US and the United Kingdom [26]. Third, externalizing and internalizing problems may be differently interpreted within each socioeconomic context. For example, in social environments characterized by violence, displaying aggressive behaviors may be considered normative or a demonstration of invulnerability to potential threats [27]. Finally, cultural norms may differentially impact how environmental factors influence mental health and suicide risk specifically. For example, although the consistent findings of sex differences in the prevalence of suicidal ideation and attempt (higher in females) and death by suicide (higher in males) in HICs have been replicated in LMICs, including South Africa, country-specific sociocultural gender norms—particularly those related to masculinity—may explain these differences in LMIC contexts [6,28–30].

The objective of this study was to investigate childhood risk factors for suicidal ideation in adolescence and young adulthood using data from the largest and longest-running birth cohort in Africa, the Birth to Twenty Plus (Bt20+) study in South Africa. Following Turecki and Brent's developmental model of suicide risk [1], a wide range of potential childhood risk factors were investigated, including early life socioeconomic adversity, ACEs, and children's externalizing and internalizing problems. These factors—considered in the model as predisposing and developmental, as opposed to precipitating, factors—were hypothesized to increase vulnerability to suicide during the life course and may be targeted by population-based suicide prevention strategies. Furthermore, we aimed to systematically document sex differences in these associations, given the known differences in the prevalence of suicide-related outcomes [30–32].

## Methods

### Study participants

This study used data from the Bt20+ cohort, a population-based longitudinal study that followed-up children from birth to adulthood [33]. The initial sample included 3,273 mothers and their singleton children born during a 7-week period in 1990 in Soweto (a historically informal settlement in Johannesburg), South Africa. Mothers were recruited from public antenatal clinics in the area in late 1989, when researchers began interviewing women who were predicted to deliver their babies during the study's enrolment period. The aim of the Bt20+ cohort was to describe the effects of rapid urbanization on the physical and psychosocial development of children during a period of dramatic political and social change in South Africa, when violence and social disorder peaked. The study is still ongoing, and the last data collection was performed in 2018 when participants were 28 years of age. In over 22 waves of data collection, the investigators assessed social and economic circumstances, family relationships, children's growth and health, schooling and employment, and mental health (S1 Fig). To overcome potential language problems, data collection was performed in the participants' language of choice (isiZulu, Sesotho, or English); consensual agreement on the phrasing of questions in the different languages was reached for each item when instrument validation was performed [34]. For this study, we analyzed data from a sample of 2,020 participants with at least 1 measure of suicidal ideation at ages 14, 17, 22, or 28 years (of the initial 3,273 participants data on suicidal ideation were not available for 1,253 participants). This analytical sample differed from the original cohort on a number of variables, including maternal age and schooling, household crowding, and assets (S1 Table). Inverse probability weighting was therefore used in all analyses to address biases due to differential attrition. Weights were derived from the independent baseline characteristics (maternal age, maternal schooling, household crowding,

and assets) predicting inclusion in the sample using a logistic regression model's individual predicted probabilities [35]. Ethical approval for the Bt20+ study was obtained from the Committee for Research on Human Subjects at the University of Witwatersrand, South Africa. Ethical clearance for the use of secondary data is only applicable where use of the data is not covered by primary data collection (the Bt20+ study) ethics approval. As this study is based on secondary data analysis within the parameters of the primary data collection's ethics clearance, no further ethical approval is necessary. All participants gave written informed consent at each data collection wave. Written consent was obtained by parents or guardians when children were minor, with verbal assent from children. From ages 16 to 28, Bt20+ participants provided written consent for each individual data collection wave.

## Assessment of suicidal ideation

Suicidal ideation was assessed at ages 14, 17, 22, and 28 years using single self-reported questions. At age 14 the item "I think about killing myself" was included in the Youth Self Report questionnaire [36]. Adolescents answered in reference to the previous 6 months using a 4-point scale (not true, sometimes true, true, and very true). These responses were dichotomized into yes (sometimes to very true) and no (not true). At ages 17 and 22, the question was asked "Have you recently found that the ideas of taking your own life kept coming into your mind?" from the General Health Questionnaire [37]. Participants answered using a 4-point scale (definitely not, I don't think so, has crossed my mind, and definitely yes), also dichotomized into yes (has crossed my mind and definitely yes) and no (otherwise). At age 28, the question "Has the thought of ending your life been on your mind?" from WHO's Self Reporting Questionnaire was asked [38]. Responses were yes or no in reference to the past 30 days. Our outcome was lifetime suicidal ideation, defined as having reported suicidal ideation at any time versus never. Follow-up intervention of care for participants was conducted at 2 points. Firstly, fieldworkers indicated if a participant appeared to be under psychological distress in study notes immediately after administration of the questionnaire, which were reviewed at the end of each day of data collection. Secondly, during data entry, a built-in calculation on items probing for "suicide ideation in the past month" gave a positive for a "mental health referral needed" variable. Participants expressing distress were accordingly put in contact with social and health services linked to the study.

## Assessment of childhood risk factors

The following childhood risk factors were investigated (see also **S1 Appendix** for the items used to assess these variables).

**Child characteristics.** Child sex (male/female), birth weight (considered as a continuous variable measured in kg following a peer-reviewer's remark; previously dichotomized into low, <2.5 kg versus nonlow, ≥2.5 kg), and birth order (first, second, third, fourth, or later born) were reported by the mother at enrollment into the study.

**Sociodemographic and maternal characteristics.** Parity (i.e., total number of pregnancies); any previous abortion (spontaneous or induced); and maternal postnatal depression were assessed when the child was 6 months of age using the Pitt Inventory [39], a validated measure consisting of 24 items (Cronbach's alpha, 0.85) assessing current feelings and changes in mood answered on a 3-point scale (yes, no, and I don't know). The inventory has been previously used in South Africa and shows good correlation with the Edinburgh Postnatal Depression Scale [40,41]. An index of socioeconomic adversity in the early life environment was created from the following variables (all reported at the time of childbirth): maternal age at childbirth; number of assets, a proxy of wealth, derived from a site-specific list (TV, fridge, car,

washing machine, and phone) according to the methodology of Filmer and Pritchett [42]; maternal education, measured as years of completed schooling; and household crowding, measured as the number of people per room living in the same household. The index was derived as the sum of poverty (<third wealth quintiles), low maternal education (≤third quartile of the distribution), low maternal age (<18 years of age at childbirth), and household overcrowding (i.e., >3 people per room) as previously described [43] and was standardized with a mean of 0 and standard deviation (SD) of 1.

**Childhood externalizing and internalizing problems.** The South African Child Assessment Schedule (SACAS) was used to ascertain child externalizing problems at ages 5, 7, and 10 years from maternal reports. The SACAS is a questionnaire based on the Child Behaviour Checklist [44] and was adapted idiomatically to the South African culture and translated into isiZulu, Sotho, and Afrikaans [40]. Externalizing problems were assessed with 35 items from the SACAS describing aggressive and rule-breaking behaviors (e.g., "Is he/she disobedient at school?" and "Does he/she physically attack people?"). Internalizing problems were assessed with 32 items from the SACAS describing anxiety and depressive symptoms (e.g., "Is he/she sad or depressed?" and "Is he/she too fearful or anxious?"). We averaged the 5-, 7-, and 10-year assessments to create the final scores that were standardized. Reliability of the SACAS has been demonstrated and validity was established in a clinical group that was shown to have significantly higher mean scores compared to a nonclinical group of children on all scales [45]. In our sample, both externalizing and internalizing problems measures showed good reliability (Cronbach's alpha were 0.80 to 0.85 and 0.76 to 0.80, respectively).

**Adverse childhood experiences.** Mothers (at child ages 5, 7, and 11) and children (at ages 11 and 13) were asked about several ACEs including experiences of material deprivation (i.e., chronic unemployment, legal problems, and chronic poverty), loss (i.e., chronic illness, disability, or death of a family member), negative family dynamics (i.e., parental substance abuse, divorce, intimate partner violence, lack of cohesion, and child separation), and child abuse and violence (i.e., child sexual and physical abuse and exposure to violence) [24]. The overall level of exposure to ACEs was computed by summing the number of reported ACEs (by either mothers or participants). This follows the cumulative risk model of ACEs, which states that it is the accumulation of adverse events, rather than the exposure to specific events, which is detrimental to health [46]. The final score was then standardized (z-score transformed).

## Data analysis

The analysis protocol was decided upon during study group meetings that took place between February and September 2020. However, there was no formal prospective analysis plan for the study. Changes to the initial planned analyses implemented following peer review were documented in the Methods section. We described continuous and categorical variables using means and SDs and counts and percentages, respectively, and used binary logistic regression to estimate the univariable associations between each childhood factor and suicidal ideation. We systematically tested the interaction between each factor and child sex and report analyses separately for males and females. Then, variables were jointly entered in a multivariable logistic regression model to estimate their independent associations with suicidal ideation, avoiding the use of collinear variables. This approach to the multivariable modeling was implemented following the comment of a reviewer. Initially, only variables that showed evidence of an association ($P < 0.05$) in the univariable analysis were used in the multivariable model. The 2 methods led to consistent results. To account for missing data in childhood factors, we estimated our model using multiple imputations: 50 imputed datasets were generated using the Amelia II package in R, which relies on a bootstrap expectation–maximization algorithm to

impute missing multivariate data [47]. Models were then estimated across all imputed datasets and results pooled. The amount of missing data is shown in **S2 Table**. Analyses were performed in R version 3.6, and the statistical significance level used was $P < 0.05$, 2 sided. This study is reported as per the Strengthening the Reporting of Observational Studies in Epidemiology (STROBE) guideline (S1 Checklist).

## Results

Among the 2,020 participants in this study, 1,042 (51.6%) were female, and 978 (48.4%) were male (**Table 1**). Approximately 11% were born of low birth weight, and 38.1% were firstborn children. Concerning mothers, 7.4% were 17 years old or younger at childbirth, and 44.6% reported living in overcrowded households.

In our sample, 469 (23.2%) participants reported suicidal ideation between the ages of 14 and 28 years. Females (lifetime estimate, 29.6%) were more likely to report suicidal ideation than males (lifetime estimate, 16.5%), with a female/male ratio of 1.7:1. As shown in **Fig 1** and **Table 2**, the prevalence of suicidal ideation increased sharply from 14 to 17 years (7.8% to 15.1%) and subsequently decreased (10.4% and 6.9% at 22 and 28 years, respectively). Although suicidal ideation was always higher in females than males, the sex gap decreased over time, and male–female rates were similar at age 28 (**Fig 2**).

**Table 3** reports the univariable associations between childhood factors and suicidal ideation. In the overall sample, female children (OR 2.09, CI 1.84 to 2.38, $P < 0.001$), those born of lower birth weight (OR for each kg lower birth weight 1.25, CI 1.11 to 1.41, $P < 0.001$), and those of higher birth order (i.e., second, third, and fourth+ born; OR for trend 1.08, CI 1.02 to 1.14, $P = 0.007$) were more likely to report suicidal ideation. Indicators of socioeconomic adversity at birth were not statistically significantly associated with suicidal ideation in the whole sample but showed evidence of a statistically significant association uniquely among males (OR 1.17, CI 1.05 to 1.29 for males; OR 0.98, CI 0.90 to 1.06 for females for each SD increase in socioeconomic adversity index; $P_{interaction} = 0.009$). This was especially the case for poverty (OR for males: 1.25, CI 1.02 to 1.52, $P = 0.031$; OR for females: 0.89, CI 0.76 to 1.04, $P = 0.152$; $P_{interaction} = 0.010$). Both externalizing (OR 1.15, CI 1.06 to 1.25, $P < 0.001$) and internalizing (OR 1.12, CI 1.04 to 1.21, $P = 0.003$) problems in childhood were significantly associated with increasing odds of suicidal ideation. In subgroup analyses, childhood externalizing problems were significantly associated with an increased likelihood of reporting suicidal ideation among both males (OR 1.28, CI 1.13 to 1.45, $P < 0.001$ for 1 SD increase in externalizing problems score) and females (OR 1.19, CI 1.06 to 1.34, $P = 0.001$; $P_{interaction} = 0.380$), while internalizing problems were significantly associated with suicidal ideation among males only (OR for males: 1.19, CI 1.05 to 1.34; $P = 0.003$; OR for females: 1.05, CI 0.95 to 1.16, $P = 0.328$), although the interaction did not reach statistical significance ($P_{interaction} = 0.118$). Finally, increased exposure to ACEs in childhood was significantly associated with higher odds of subsequently reporting suicidal ideation (OR 1.10, CI 1.03 to 1.18, $P = 0.003$ for 1 SD increase in ACEs score) and stratified analyses by sex suggested that the association was stronger among females (OR 1.14, CI 1.05 to 1.24, $P = 0.003$) than males (OR 1.06, CI 0.95 to 1.18, $P = 0.299$), although the interaction was not statistically significant ($P_{interaction} = 0.308$).

Multivariable analyses (**Table 4**) showed that socioeconomic adversity (OR 1.13, CI 1.02 to 1.25, $P = 0.031$) and externalizing problems (OR 1.22, CI 1.08 to 1.39, $P = 0.002$) were independently associated with an increased likelihood of reporting suicidal ideation among males, while internalizing problems only approached statistical significance (OR 1.12, CI 0.99 to 1.27, $P = 0.083$). Among females, independent factors significantly associated with suicidal ideation were lower birth weight (OR 1.20, CI 1.02 to 1.41, $P = 0.030$), ACEs (OR 1.11, CI 1.01 to 1.21,

**Table 1. Sociodemographic characteristics of the sample.**

| | Whole sample (N = 2,020) | Males (n = 978) | Females (n = 1,042) | P |
|---|---|---|---|---|
| Female sex | 1,042 (51.6) | | | |
| Low birth weight | 219 (10.9) | 94 (9.6) | 125 (12.0) | 0.102 |
| Birth order | | | | 0.308 |
| First | 769 (38.1) | 363 (37.1) | 406 (39.0) | |
| Second | 584 (28.9) | 284 (29.0) | 300 (28.8) | |
| Third | 352 (17.4) | 164 (16.8) | 188 (18.0) | |
| Fourth+ | 315 (15.6) | 167 (17.1) | 148 (14.2) | |
| Mean* | 2.11 (1.08) | 2.14 (1.10) | 2.07 (1.06) | 0.189 |
| Socioeconomic adversity index (z-score)* | 0.00 (1.00) | 0.05 (1.01) | −0.04 (0.99) | 0.064 |
| Low maternal age | 149 (7.4) | 68 (7.0) | 81 (7.8) | 0.527 |
| Low maternal education | 1,096 (59.1) | 546 (61.1) | 550 (57.3) | 0.102 |
| Household crowding | 772 (44.6) | 384 (46.0) | 388 (43.4) | 0.292 |
| Poverty | 890 (44.1) | 449 (45.9) | 441 (42.3) | 0.115 |
| Cumulative ACE score (z-score)*[a] | 0.00 (1.00) | −0.01 (1.01) | 0.01 (0.99) | 0.711 |
| Postnatal maternal depression | 199 (17.2) | 106 (18.4) | 93 (16.0) | 0.316 |
| Parity | | | | 0.446 |
| First | 769 (38.1) | 363 (37.1) | 406 (39.0) | |
| Second | 584 (28.9) | 284 (29.0) | 300 (28.8) | |
| Third | 352 (17.4) | 164 (16.8) | 188 (18.0) | |
| Fourth | 165 (8.2) | 86 (8.8) | 79 (7.6) | |
| Fifth+ | 150 (7.4) | 81 (8.3) | 69 (6.6) | |
| Mean* | 2.18 (1.23) | 2.22 (1.26) | 2.14 (1.20) | 0.146 |
| Previous abortions/stillbirths | 219 (10.8) | 122 (12.5) | 97 (9.3) | 0.027 |
| Externalizing problems (z-score)* | 0.00 (1.00) | −0.03 (1.00) | 0.05 (0.97) | <0.001 |
| Internalizing problems (z-score)* | 0.00 (1.00) | 0.23 (1.02) | −0.05 (0.89) | 0.087 |

Counts and %, except for continuous variables (*), described as mean and SD.

[a]Unstandardized values for the whole sample, males, and females are 3.00 (2.07), 2.93 (2.03), and 3.05 (2.10), respectively. All tests are 2 sided and considered statistically significant at P < 0.05. P values have been obtained using Student t tests and chi-squared tests for continuous and categorical variables, respectively.

ACE, adverse childhood experience; SD, standard deviation.

P = 0.030), higher birth order (for each birth order increase, OR 1.14, CI 1.06 to 1.23, P < 0.001), and externalizing problems (OR 1.16, CI 1.03 to 1.31, P = 0.011). Multivariable analysis including only factors associated with suicidal ideation at P < 0.05 in the univariable analyses yielded consistent results (S3 Table).

## Discussion

This study described the prevalence of suicidal ideation from ages 14 to 28 and identified key socioeconomic and individual-level childhood factors associated with suicidal ideation among South African youth. We found that, overall, 23.2% of participants reported suicidal ideation, with a peak in prevalence at age 17 years and an overall 1.7:1 female/male ratio. Females reported significantly higher rates than males at all time points except for age 28 when rates for males and females were similar. We found that males exposed to socioeconomic adversity and those who experienced externalizing problems in childhood were more likely to consider ending their lives during their teenage years and early adulthood. Among females, childhood factors associated with suicidal ideation included lower birth weight, ACEs, higher birth order, and externalizing problems.

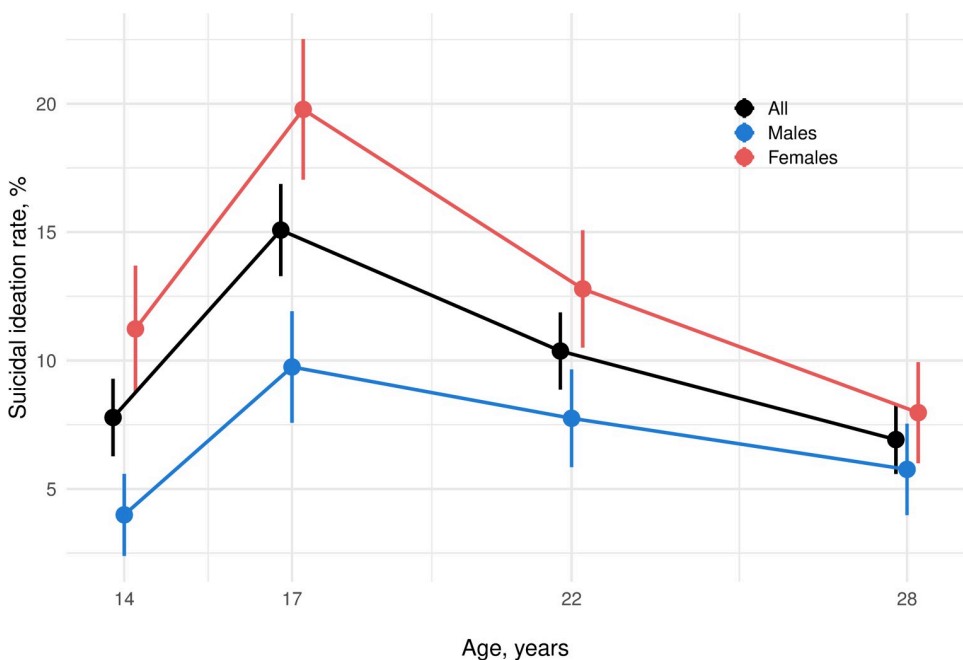

**Fig 1. Prevalence of suicidal ideation in the Bt20+ cohort.** The figure illustrates the prevalence of suicidal ideation at each assessment occasion and their change over time. Percentages are relative to the individuals in the cohort for which suicidal ideation data were available at the given age, namely 1,209 (632 females and 577 males), 1,532 (814 females and 718 males), 1,582 (821 females and 761 males), and 1,388 (728 females and 660 males) participants at ages 14, 17, 22, and 28 years, respectively. Vertical bars represent 95% CIs around the point prevalence. Bt20+, Birth to Twenty Plus.

## Added knowledge to existing research

To the best of our knowledge, this is the first study to prospectively examine the prevalence and childhood risk factors of suicidal ideation across adolescence and young adulthood among youth in sub-Saharan Africa. As most previous longitudinal studies on the topic are based on samples from HICs and countries outside the African continent, our findings add to the current literature about the prevalence and etiology of suicidal ideation in LMICs in Africa that can be used by policymakers to elaborate local suicide prevention strategies.

**Table 2. Prevalence rates of suicidal ideation in the Bt20+ cohort.**

|  | Whole sample | | By sex | | | | | |
|---|---|---|---|---|---|---|---|---|
|  |  |  | Females | | Males | | Sex comparison | |
| Age | n/N | % (95% CI) | n/N | % (95% CI) | n/N | % (95% CI) | RR (95% CI) | P |
| 14 | 94/1,209 | 7.8 (6.3 to 9.3) | 71/632 | 11.2 (8.8 to 13.7) | 23/577 | 4.0 (2.4 to 5.6) | 2.81 (1.79 to 4.61) | <0.001 |
| 17 | 231/1,532 | 15.1 (13.3 to 16.9) | 161/814 | 19.8 (17.0 to 22.5) | 70/718 | 9.8 (7.6 to 11.9) | 2.03 (1.54 to 2.70) | <0.001 |
| 22 | 164/1,582 | 10.4 (8.9 to 11.9) | 105/821 | 12.8 (10.5 to 15.1) | 59/761 | 7.8 (5.9 to 9.6) | 1.65 (1.20 to 2.28) | 0.001 |
| 28 | 96/1,388 | 6.9 (5.6 to 8.3) | 58/728 | 8.00 (6.0 to 9.9) | 38/660 | 5.8 (4.0 to 7.5) | 1.38 (0.92 to 2.10) | 0.130 |

The table shows the number of participants reporting suicide attempt (n) relative to the number of assessed participants (N) at any assessment age, as well as the rate (%) with 95% CI. Statistics are provided for the whole sample and for females and males separately. The prevalence of suicidal ideation in females versus males is compared using RR with 95% CI, and P values were computed using chi-squared tests. All tests are 2 sided and considered statistically significant at P < 0.05.

Bt20+, Birth to Twenty Plus; CI, confidence interval; RR, risk ratio.

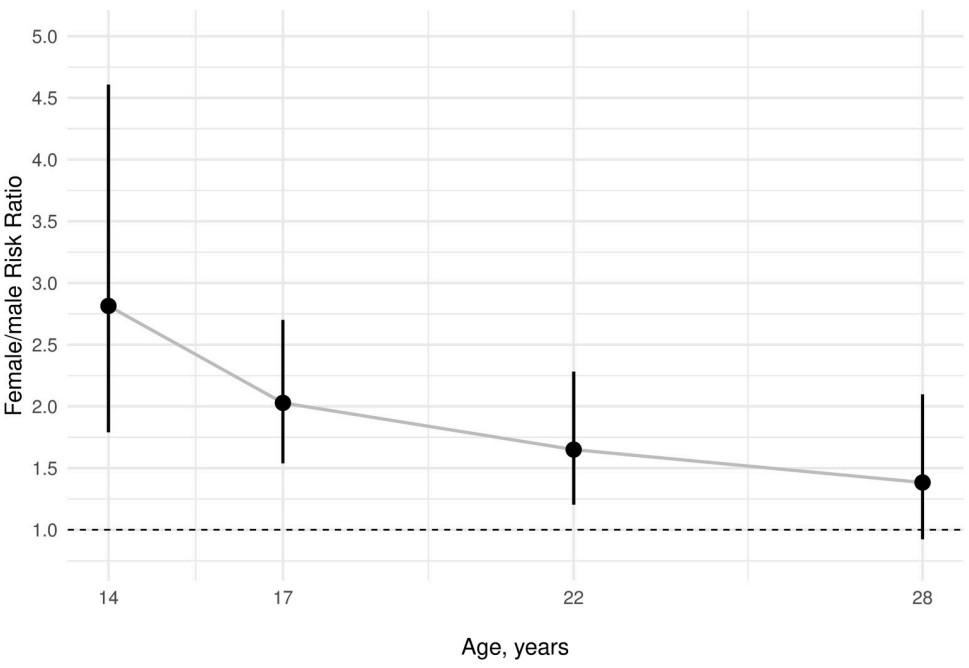

**Fig 2. Female-to-male RR for suicidal ideation at each age.** The figure represents the comparisons of suicidal ideation rates in females versus males as risk ratios (RR) at each age. RR were calculated by dividing the prevalence rate in females by the prevalence rate in males, e.g., for age 14 years: (71/632)/(23/577) = 2.81. Vertical bars represent 95% CIs of the RR. The horizontal dashed line represents the null (RR = 1), and CI bars indicates that the RR is not statistically significant at $P < 0.05$, except at age 28 years. RR, risk ratio.

## Comparison of the findings with previous research

Our findings of the prevalence of suicidal ideation, the sex distribution, and the peak at age 17 years are in line with previous LMIC and South African prevalence studies that have reported increased rates of suicidal ideation in later adolescence (17.8%; 15 to 17 years) compared to earlier adolescence (15.9%; 13 to 14 years) [6,48] and higher rates in females (8.5%; 10 to 18 years) compared to males (5.6%; 10 to 18 years) [29,48,49]. Consistently, previous meta-analyses of LMIC studies reported a higher prevalence of suicidal ideation, planning, and attempt among female youth, with those in the African region reporting the highest rates [6,31]. Our findings on the risk factors for suicidal ideation are also consistent with prior cross-sectional studies in Africa. For example, a previous investigation among young South African men also found an association between poverty and past month suicidal ideation [50]. In another South African study, ACEs measured at baseline (10 to 18 years) were significantly associated with suicidal ideation among adolescents 1 year later [29]. This study also reported a higher prevalence of suicidal ideation among females compared to males but did not examine whether the association between ACEs and suicidal ideation differed by sex. These findings are also consistent with those from studies conducted in HICs that reported associations between poverty and ACEs in childhood and suicide-related outcomes over the life span [51,52], suggesting that such factors are deleterious for human development and mental health irrespective of cultural and socioeconomic variations. However, studies examining sex differences in the association between childhood factors and suicidal outcomes have found a different pattern of results from ours. For example, a recent meta-analysis of longitudinal studies (mostly from HICs) reported no sex differences in the association of ACEs and internalizing problems with suicide attempt and death by suicide, but found that externalizing problems were only associated with

**Table 3. Univariable associations between childhood factors and suicidal ideation.**

| | Whole sample | | Males | | Females | | Interaction |
|---|---|---|---|---|---|---|---|
| | OR (CI) | *P* | OR (CI) | *P* | OR (CI) | *P* | *P* |
| **Child characteristics** | | | | | | | |
| Female sex | 2.09 (1.84 to 2.38) | <0.001 | - | | - | | - |
| Birth weight (each kg decrease) | 1.25 (1.11 to 1.41) | <0.001 | 1.19 (0.98 to 1.43) | 0.075 | 1.18 (1.01 to 1.39) | 0.048 | 0.931 |
| Birth order | | | | | | | |
| Second | 1.11 (0.96 to 1.29) | 0.172 | 1.00 (0.78 to 1.28) | 0.992 | 1.20 (0.99 to 1.46) | | |
| Third | 0.86 (0.71 to 1.04) | 0.119 | 0.78 (0.57 to 1.07) | 0.125 | 0.90 (0.71 to 1.15) | | |
| Fourth+ | 1.45 (1.21 to 1.73) | <0.001 | 1.26 (0.95 to 1.66) | 0.107 | 1.79 (1.41 to 2.27) | | |
| Trend | 1.08 (1.02 to 1.14) | 0.007 | 1.04 (0.95 to 1.14) | 0.429 | 1.14 (1.06 to 1.22) | 0.001 | 0.120 |
| **Sociodemographic and maternal characteristics** | | | | | | | |
| Socioeconomic adversity index | 1.03 (0.97 to 1.10) | 0.355 | 1.17 (1.05 to 1.29) | 0.003 | 0.98 (0.90 to 1.06) | 0.576 | 0.009 |
| Low maternal age | 0.95 (0.78 to 1.17) | 0.654 | 0.97 (0.69 to 1.37) | 0.868 | 0.92 (0.70 to 1.19) | 0.508 | 0.788 |
| Low maternal education | 1.11 (0.97 to 1.26) | 0.132 | 1.31 (1.04 to 1.64) | 0.023 | 1.06 (0.89 to 1.25) | 0.514 | 0.150 |
| Household crowding | 1.06 (0.92 to 1.23) | 0.404 | 1.24 (0.98 to 1.57) | 0.068 | 0.99 (0.82 to 1.19) | 0.917 | 0.137 |
| Poverty | 0.99 (0.87 to 1.12) | 0.840 | 1.25 (1.02 to 1.52) | 0.031 | 0.89 (0.76 to 1.04) | 0.152 | 0.010 |
| Postnatal maternal depression | 1.17 (0.89 to 1.54) | 0.216 | 1.30 (0.86 to 1.96) | 0.235 | 1.15 (0.81 to 1.62) | 0.247 | 0.628 |
| Parity | | | | | | | |
| Second | 1.11 (0.96 to 1.29) | 0.172 | 1.00 (0.78 to 1.28) | 0.992 | 1.20 (0.99 to 1.46) | | |
| Third | 0.86 (0.71 to 1.04) | 0.119 | 0.78 (0.57 to 1.07) | 0.125 | 0.90 (0.71 to 1.15) | | |
| Fourth | 1.44 (1.14 to 1.81) | 0.002 | 1.26 (0.88 to 1.80) | 0.214 | 1.73 (1.28 to 2.35) | | |
| Fifth+ | 1.46 (1.15 to 1.85) | 0.002 | 1.26 (0.87 to 1.82) | 0.216 | 1.86 (1.35 to 2.58) | | |
| Trend | 1.08 (1.03 to 1.13) | 0.003 | 1.04 (0.96 to 1.13) | 0.322 | 1.13 (1.06 to 1.21) | <0.001 | 0.106 |
| Previous abortions/stillbirths | 0.94 (0.77 to 1.16) | 0.576 | 0.92 (0.67 to 1.27) | 0.615 | 1.09 (0.82 to 1.45) | 0.549 | 0.439 |
| **Childhood mental health** | | | | | | | |
| Externalizing problems | 1.15 (1.06 to 1.25) | <0.001 | 1.28 (1.13 to 1.45) | <0.001 | 1.19 (1.06 to 1.34) | 0.001 | 0.380 |
| Internalizing problems | 1.12 (1.04 to 1.21) | 0.003 | 1.19 (1.05 to 1.34) | 0.003 | 1.05 (0.95 to 1.16) | 0.328 | 0.118 |
| **ACEs** | | | | | | | |
| Cumulative ACE score | 1.10 (1.03 to 1.18) | 0.003 | 1.06 (0.95 to 1.18) | 0.299 | 1.14 (1.05 to 1.24) | 0.003 | 0.308 |
| Material deprivation | 1.16 (1.08 to 1.24) | <0.001 | 1.04 (0.93 to 1.16) | 0.535 | 1.26 (1.16 to 1.38) | <0.001 | 0.007 |
| Loss | 1.04 (0.98 to 1.11) | 0.219 | 1.02 (0.91 to 1.13) | 0.789 | 1.05 (0.97 to 1.14) | 0.233 | 0.609 |
| Family dynamics | 1.06 (1.00 to 1.13) | 0.057 | 1.09 (0.98 to 1.21) | 0.096 | 1.04 (0.95 to 1.12) | 0.403 | 0.430 |
| Abuse and violence | 1.01 (0.94 to 1.09) | 0.796 | 0.99 (0.87 to 1.12) | 0.858 | 1.05 (0.95 to 1.15) | 0.334 | 0.474 |

All tests are 2 sided and considered statistically significant at *P* < 0.05. *P* values have been obtained using univariable logistic regressions.

ACE, adverse childhood experience; OR, odds ratio.

suicide attempt in males [32]. Our finding of an association between externalizing problems and suicidal ideation among both males and females could be explained by previous findings of a higher prevalence of childhood externalizing problems in LMICs compared to HICs among both males and females [27,53]. Importantly, while internalizing problems are often associated with suicidal ideation in HICs [3,54], they were not independent predictors in our study. This is consistent with previous studies in HICs that have shown that externalizing behavior and comorbid internalizing–externalizing behaviors are more strongly associated with suicide-related outcomes than internalizing behavior alone [12,14,15]. While other studies found independent associations of internalizing behavior with suicide-related outcomes, these studies often measured internalizing behavior in adolescence rather than in childhood [54,55]. However, it is also important to consider that the lack of association for internalizing

**Table 4. Multivariable associations between childhood factors and suicidal ideation.**

|  | Males | | Females | |
| --- | --- | --- | --- | --- |
|  | OR (CI) | *P* | OR (CI) | *P* |
| Birth weight (each kg decrease) | 1.15 (0.95 to 1.41) | 0.146 | 1.20 (1.02 to 1.41) | 0.030 |
| Birth order (trend) | 1.02 (0.93 to 1.12) | 0.642 | 1.15 (1.07 to 1.24) | <0.001 |
| Any previous abortion | 0.92 (0.67 to 1.27) | 0.616 | 0.97 (0.73 to 1.30) | 0.848 |
| Postnatal maternal depression | 1.01 (0.97 to 1.06) | 0.576 | 1.02 (0.98 to 1.06) | 0.441 |
| Socioeconomic adversity score | 1.13 (1.01 to 1.26) | 0.031 | 0.94 (0.86 to 1.02) | 0.132 |
| Externalizing problems | 1.23 (1.08 to 1.40) | 0.002 | 1.16 (1.03 to 1.30) | 0.011 |
| Internalizing problems | 1.12 (0.99 to 1.27) | 0.083 | 1.00 (0.90 to 1.11) | 0.939 |
| ACE score | 0.97 (0.86 to 1.09) | 0.584 | 1.11 (1.01 to 1.21) | 0.030 |

All tests are 2 sided and considered statistically significant at *P* < 0.05. *P* values have been obtained using multivariable logistic regressions.

ACE, adverse childhood experience; OR, odds ratio.

problems may be due to the developmental age in which internalizing problems have been measured, since some previous studies in HICs suggested associations between internalizing problems measured in adolescence and suicide-related outcomes. Finally, different from studies in HICs [56,57], we did not find an association between maternal postnatal depression and offspring suicidal ideation. However, previous Bt20+ studies have found that maternal postnatal depression is associated with offspring internalizing problems in both childhood [40] and early adulthood [58]. Therefore, the absence of associations with suicidal ideation in the present study may indicate that, although maternal depression increases risk for offspring internalizing problems, it is not sufficient to differentiate participants who consider suicide from those who do not in the specific socioeconomic context of South Africa. However, maternal depression had a high rate of missing data in our sample (**S1 Table**). Therefore, although maternal depression was not associated with attrition, caution should be used to interpret this lack of association, as the mothers with the highest depressive symptoms may have dropped out from the study.

## Strengths and limitations

A key strength of this paper is the use of longitudinal data from the longest birth cohort in Africa to prospectively examine the association of the childhood environment with suicidal ideation in youth. Assessments relied on validated measures and care was taken to adapt them to the local setting. Additionally, the majority of childhood factors were reported by mothers while suicidal ideation was self-reported by offspring, thus reducing bias due to shared method variance. The longitudinal nature of this study also enabled us to assess risk factors multiple times throughout participants' childhood. In addition to these strengths, certain limitations of the study should also be acknowledged. Attrition was substantial, although comparable with other longitudinal cohorts [59]. We used inverse probability weighting to address differential attrition, which may partially account for related biases. Furthermore, due to differential attrition [33,60], the majority of our sample was Black South African, the overwhelmingly largest group in the country. There is some evidence suggesting that prevalence rates of death by suicide and suicidal ideation are higher in population groups not represented in our sample [49]; thus, our results may not be generalizable to South African youth from minority population groups. Another limitation was the use of 3 different questionnaires to assess suicidal ideation. Although the items used assessed the same construct, broadly defined as having thought about ending one's life and was appropriate to the life stage of the sample, measurement differences

may have introduced bias. This limitation is due to the fact that the Bt20+ cohort was not initially designed to study specific suicide-related outcomes. However, it worth noting that (1) this study considered "any suicidal ideation" as an outcome rather than a measure of its intensity, thus reducing bias arising from using different instruments; and (2) bias would lead to underestimation of our associations, leading to conservative results. Moreover, although suicidal ideation is a key aspect of suicide risk, this study did not measure other important aspects such as suicide attempt and transition from ideation to attempt. Another limitation is that some of the childhood factors were only measured once, and our analysis did not take into account their time-varying nature (e.g., socioeconomic adversity). Furthermore, only parent reports of child behavior were available in early and middle childhood. This may have introduced measurement errors, especially for internalizing behavior measures that can be affected by maternal state of mind and attitudes. Finally, the childhood factors investigated in our study reflect a conceptual model that mostly emphasizes risk factors rather than protective and mitigating factors. To enhance suicide prevention, further research is needed to understand the factors decreasing and buffering suicide risk, especially for children exposed to socioeconomic adversity and those exhibiting behavioral problems.

## Implications and next steps for research, clinical practice, and public policy

Given that the African region bears the heaviest burden of suicide-related outcomes among youth, further studies across the continent are needed to understand how country-specific factors impact the association between the childhood environment and suicide-related outcomes and to explore sex differences in these associations to inform prevention. Further qualitative studies are also needed to complement epidemiological studies by unpacking how childhood risk factors and sociodemographic characteristics are associated with suicidal ideation in the sociocultural contexts of countries in the African region [61]. For example, the handful of qualitative studies conducted in South Africa [30,61–65] highlight societal expectations of masculinity (e.g., escaping from situations in which they were unable to live up to traditional understandings of masculinity) and protest against masculine dominance (e.g., for females, experiencing intimate partner violence) as important factors for suicidal behavior. These factors play a role in the sociocultural context within which suicidal behaviors occur and how context can inform interventions [66]. Our findings also emphasize the need to consider how individual and social factors interact in increasing risk of suicide-related outcomes. Prior studies point to the high rates of poverty and unemployment in South Africa and their relation to a breakdown in family life, which can result in a number of social problems including child abuse and neglect [65]. Inability to cope with such adverse socioeconomic experiences could result in psychological distress, which is an important risk factor for suicide.

## Conclusions

In this longitudinal study on the prevalence and etiology of suicidal ideation in South African youth, we found that prevalence rates peaked at age 17 and decreased continuously until age 28. Prevalence rates were higher among females than males, and we found sex differences in the association of childhood individual, familial, and environmental factors with youth suicidal ideation. As these factors (e.g., externalizing problems, socioeconomic adversity, and ACEs) are highly prevalent in South Africa, our findings support the need for a population-based approach to suicide prevention aiming at reducing the pervasiveness of childhood adversity and increasing societal well-being. Given the high burden of suicide-related outcomes in LMICs, especially in the African region, further quantitative and qualitative research is needed

to understand how country-specific factors influence suicide-related outcomes. Such research can inform the development of interventions to reduce the burden of youth suicide.

## Disclaimers

Opinions expressed and conclusions arrived at are those of the authors and are not to be attributed to the CoE in Human Development.

## Supporting information

**S1 Checklist. STROBE Checklist.** STROBE, Strengthening the Reporting of Observational Studies in Epidemiology.
(DOCX)

**S1 Appendix. Questionnaires.**
(PDF)

**S1 Table. Comparison of the characteristics of participants included and not included in the analysis sample.**
(DOCX)

**S2 Table. Count and proportion of missing data in the analysis variable.**
(DOCX)

**S3 Table. Multivariable associations between childhood factors and suicidal ideation using factors associated with suicidal ideation in the univariable analysis at $P < 0.05$.**
(DOCX)

**S1 Fig. Representation of the Bt20+ cohort assessments used in this investigation.** Bt20+, Birth to Twenty Plus.
(DOCX)

## Acknowledgments

MNA was funded by a Vanier Canada Graduate Scholarship from Canada's Social Sciences and Humanities Research Council. SB is a CIFAR Azrieli Global Scholar in the Brain, Mind and Consciousness Program. The support of the DSI-NRF Centre of Excellence (CoE) in Human Development at the University of the Witwatersrand, Johannesburg in the Republic of South Africa toward this research is hereby acknowledged. We thank the Birth to Twenty study team and all participants who enable this research to continue. The Bt20+ sample has been funded by the Wellcome Trust, the South African Medical Research Council, the Human Sciences Research Council, the University of the Witwatersrand, and the Bill & Melinda Gates Foundation among others.

## Author Contributions

**Conceptualization:** Massimiliano Orri, Marilyn N. Ahun, Sara Naicker, Sahba Besharati, Linda M. Richter.

**Data curation:** Linda M. Richter.

**Formal analysis:** Massimiliano Orri.

**Funding acquisition:** Massimiliano Orri, Linda M. Richter.

**Investigation:** Linda M. Richter.

**Methodology:** Massimiliano Orri, Marilyn N. Ahun, Sara Naicker, Sahba Besharati, Linda M. Richter.

**Resources:** Linda M. Richter.

**Supervision:** Linda M. Richter.

**Visualization:** Massimiliano Orri.

**Writing – original draft:** Massimiliano Orri, Marilyn N. Ahun.

**Writing – review & editing:** Massimiliano Orri, Marilyn N. Ahun, Sara Naicker, Sahba Besharati, Linda M. Richter.

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
