## [Editor Report · Decision Letter 0]

12 Aug 2021

Dear Dr Orri, 

Thank you for submitting your manuscript entitled "Childhood factors associated with suicidal ideation among South African youth A 28-year longitudinal study using the Birth to Twenty Plus cohort" for consideration by PLOS Medicine.

Your manuscript has now been evaluated by the PLOS Medicine editorial staff as well and I am writing to let you know that we would like to send your submission out for external peer review.

Please re-submit your manuscript within two working days, i.e. by Aug 16 2021 11:59PM.

Kind regards,

Caitlin Moyer, Ph.D.

Associate Editor

PLOS Medicine

---

## [Decision Letter · Decision Letter 1]

30 Nov 2021

Dear Dr. Orri,

Thank you very much for submitting your manuscript "Childhood factors associated with suicidal ideation among South African youth A 28-year longitudinal study using the Birth to Twenty Plus cohort" (PMEDICINE-D-21-03460R1) for consideration at PLOS Medicine. 

Your paper was evaluated by a senior editor and discussed among all the editors here. It was also discussed with an academic editor with relevant expertise, and sent to four independent reviewers, including a statistical reviewer. The reviews are appended at the bottom of this email and any accompanying reviewer attachments can be seen via the link below:

[LINK]

In light of these reviews, I am afraid that we will not be able to accept the manuscript for publication in the journal in its current form, but we would like to consider a revised version that addresses the reviewers' and editors' comments. Obviously we cannot make any decision about publication until we have seen the revised manuscript and your response, and we plan to seek re-review by one or more of the reviewers. 

We expect to receive your revised manuscript by Dec 21 2021 11:59PM. Please email us (plosmedicine@plos.org) if you have any questions or concerns.

We look forward to receiving your revised manuscript. 

Sincerely,

Caitlin Moyer, Ph.D.

Associate Editor

PLOS Medicine

plosmedicine.org

Comments from the Academic Editor:

1. The authors have selectively focused on adversity and not on potentially protective or mitigating factors e.g. relating to schooling/educational success, social capital/extended family indicators (if measured). This appears to be a limitation.

2. Was there any conceptual model that drove the selection the individual, familial and environmental exposures? for example, was substance use considered as a factor?

3. is it reasonable to expect that predictors of suicidal ideation at those different time-points would be the same?

4. It is surprising that postnatal depression is not a predictor. What could be the reason for that?

5. I agree with the statistical reviewer's concerns about the way that variables were selected for inclusion into the multivariable analysis, and also the interpretation of interactions.

Other editorial points:

6. Data availability statement: Thank you for providing the link to request data access. Please clarify if there is a more direct link available (e.g. https://www.wits.ac.za/coe-human/open-access-datasets/). If possible please provide a (non-author) contact email address to which data inquiries can be directed.

According to this website, the following text should be included in the acknowledgements: “The support of the DSI-NRF Centre of Excellence in Human Development at the University of the Witwatersrand, Johannesburg in the Republic of South Africa towards this research is hereby acknowledged. Opinions expressed and conclusions arrived at, are those of the author and are not to be attributed to the CoE in Human Development.” It seems only part of this statement was included.

7. Throughout: Please include line numbers running throughout the text with the revised version.

8. Abstract: Please combine the Methods and Findings sections into one section, “Methods and findings”.

9. Abstract: Methods: We suggest emphasizing early on that the primary outcome of interest was suicidal ideation reported at any age.

10. Abstract: Methods and Findings: Please quantify the main results (with 95% CIs and p values). Please mention the important dependent variables that are adjusted for in the analyses.

11. Abstract: Methods and Findings: Please clarify “high birth order” briefly.

12. Abstract: Methods and Findings: In the last sentence of the Abstract Methods and Findings section, please describe the main limitation(s) of the study's methodology.

13. Abstract: Conclusions: Please address the study implications without overreaching what can be concluded from the data; the phrase "In this study, we observed ..." may be useful.

14. Author summary: At this stage, we ask that you include a short, non-technical Author Summary of your research to make findings accessible to a wide audience that includes both scientists and non-scientists. The Author Summary should immediately follow the Abstract in your revised manuscript. This text is subject to editorial change and should be distinct from the scientific abstract. Please see our author guidelines for more information: https://journals.plos.org/plosmedicine/s/revising-your-manuscript#loc-author-summary

15. Throughout: Please use square brackets for in-text citations, rather than superscript numbers. Please place the reference before the sentence punctuation, and please do not include spaces within brackets where multiple references are indicated, for example [1,2].

16. Methods: Please ensure that the study is reported according to the STROBE guideline, and include the completed STROBE checklist as Supporting Information. Please add the following statement, or similar, to the Methods: "This study is reported as per the Strengthening the Reporting of Observational Studies in Epidemiology (STROBE) guideline (S1 Checklist)."

17. Methods: Did your study have a prospective protocol or analysis plan? Please state this (either way) early in the Methods section.

18. Methods: Please provide some brief details on how mothers were recruited for the Bt20+ cohort.

19. Methods: Please clarify the information on phrasing of questions across languages: “Data

collection was performed mainly in isiZulu, Sesotho, or English, with consensual agreement on the

phrasing of questions asked across the different languages.”

20. Methods: “For this study, we analysed data from a sample of 2,020 participants with at least one measure of suicidal ideation at ages 14, 17, 22, or 28 years. This analytical sample differed from the original cohort on a number of variables, including maternal age and schooling, household crowding, and assets. Inverse probability weighting was therefore used in all analyses to address biases due to differential attrition.”

Please provide a table comparing the sample of 2,020 participants included here and the original cohort. A participant flowchart would be helpful.

21. Ethical approval: Please clarify if this report is a secondary analysis of data collected from the Bt20+ study and if ethical approval for this analysis was waived.

Please clarify the nature of written consent from all participants at all waves- was written consent obtained from parents/guardians with written assent from the children, with written consent provide at waves conducted during adulthood?

Please also note whether and how individuals indicating having experienced suicidal ideation were followed up for additional intervention and care.

22. Methods: Study waves at age 17 and 22: Please clarify if the response of “No more than usual” to “Have you recently found that the ideas of taking your own life kept coming into your mind?” was dichotomized as yes or no, and if categorized as “no” please explain as this seems as if it could indicate suicidal ideation.

23. Methods: It would be helpful to include copies of the relevant questions used to obtain the primary outcome and childhood risk factor data, as supporting information.

24. Methods: Please describe the inverse probability weighting that was used to account for study attrition. Please include more description of the multiple imputation methods. It would be helpful to indicate the amount of missing responses for each relevant outcome/factor within a Table.

25. Methods: Please specify the significance level used (e.g., P<0.05, two-sided).

26. Results: Please provide an analysis comparing characteristics of those for which data were missing compared with those included in the study.

27. Results: For all results presented in the text (e.g from univariable and multivariable analyses) please present both the 95% CIs and p values. Please present the overall associations for each factor. If results stratified by sex are presented, please present the findings for both sexes, and please also report the interaction.

28. Results: “This was especially true for poverty (1.25, CI 1.02- 1.52), low maternal education (OR 1.31, CI 1.04-1.64), and household crowding (OR 1.24, CI 0.98-1.57).” Please clarify that this describes the associations for males.

29. Results: Childhood internalizing problems: For differences reported between males and females, it seems as if the interaction by sex did not reach statistical significance. Please mention in the text that while the association with internalizing problems was significant for males (and please report the result for females) there was no evidence to support an interaction effect by sex. Please present the overall associations for externalizing and internalizing problems in addition to the sex-stratified analyses.

30. Discussion: Please present and organize the Discussion as follows: a short, clear summary of the article's findings; what the study adds to existing research and where and why the results may differ from previous research; strengths and limitations of the study; implications and next steps for research, clinical practice, and/or public policy; one-paragraph conclusion.

31. Discussion: Please temper statements related to primacy with “To the best of our knowledge” or similar: “This is the first study to prospectively examine associations…”

32. Discussion: “A key strength of this paper is the use of longitudinal data from the longest birth cohort in Africa to prospectively examine the impact of the early childhood environment on suicidal ideation in youth.” Here and throughout, please revise to avoid language that implies causality. Instead, please refer to associations.

33. Figure 1: Please present this information in a table format. Please note the numbers as well as percentages reported for both males and females. Please indicate in the legend what is represented by the “error bars” and please note that the percentages are relative to the individuals in the cohort for which suicidal ideation data were available at the given age.

34. Figure 2: Please provide a descriptive legend describing the figure, including how risk ratio was determined and the meaning of the points and bars.

35. Table 1: Please specify the significance level used (eg, P<0.05, two-sided) and the statistical test used to derive the p value.

36. Table 2: Please provide the p values for the associations for the whole sample, males, and females. In the legend, please note the statistical tests used.

37. Table 3: Please provide the p values for these associations. In the legend, please note the statistical tests used.

38. References: Please use the "Vancouver" style for reference formatting, and see our website for other reference guidelines https://journals.plos.org/plosmedicine/s/submission-guidelines#loc-references

Comments from the reviewers:

Reviewer #1: I confine my remarks to statistical aspects of this paper. The general approach is fine, but I have a few issues to resolve before I can recommend publication.

Abstract (and similar in main text) - when comparing males to females be careful to not accept the null. You can either insert a bunch of "significantly" or else, rather than say "not associated" say things like "higher in males" or "higher in females".

p. 3 I'm not sure you can say that preventing suicidal ideation would enhance mental well being. It could be the other way around. 

 "seriously considering and attempting suicide" - which one? Either?

p. 6 Don't categorize birth weight and (maybe) don't categorize depression. Categorizing a continuous variable is nearly always a bad idea. See my blog post https://medium.com/@peterflom/what-happens-when-we-categorize-an-independent-variable-in-regression-77d4c5862b6c

p. 7 This method of model building is known as bivariate screening and it is not recommended. All the results will be wrong. P values are too low, standard errors are too small, parameter estimates are biased away from 0. See Harrell, Regression Modeling Strategies.

Peter Flom 

Reviewer #2: Title: Childhood factors associated with suicidal ideation among South African youth A 28-year longitudinal study using the Birth to Twenty Plus cohort

* What are the main claims of the paper and how significant are they for the discipline? 

This is a well-written manuscript focusing on an important subject in child mental health. The research aims to show the possible impact of individual, familial and environmental factors on the suicidal ideation of young people in South Africa taken from a cohort sample. 

* Are the claims properly placed in the context of the previous literature? Have the authors treated the literature fairly? 

Discussion was good and comprehensive. 

* Do the data and analyses fully support the claims? If not, what other evidence is required? 

Overall, the sample and methods of the study are robust. However, could the authors explain why they have not used ACE-IQ to find out about adverse childhood experiences? Furthermore, why did they not ask for ACEs after age 13? I believe this might have changed the cumulative ACE score and therefore its possible association with suicidal ideation. What is the mean ACE score for the sample? Why did the authors use cumulative ACE score rather than the mean score which is more commonly used in the field? Have the authors looked into the impacts of individual ACE components on suicidal ideation?

The authors also mention about attrition being substantial despite being comparable with other cohorts. Have the authors compared baseline characteristics of participants who were lost to follow-up with clinical characteristics of those remaining? If so, what are the results, and could they have an effect on the results of the paper?

* PLOS Medicine encourages authors to publish detailed methods as supporting information online. Do any particular methods used in the manuscript warrant such publication? If a protocol is already provided, for example for a randomized controlled trial, are there any important deviations from it? If so, have the authors explained adequately why the deviations occurred?

I have not seen any information about this. 

* Is this paper outstanding in its discipline? If yes, what makes it outstanding? If not, why not?

It is an important paper taken from a cohort sample, looking into an important subject in mental health, and aiming to find out the factors associated with suicidal ideation in a population from a low/middle income country.

* Does the study conform to any relevant guidelines such as CONSORT, MIAME, QUORUM, STROBE, and the Fort Lauderdale agreement?

N/A

* Are details of the methodology sufficient to allow the experiments?

Yes. 

* Is any software created by the authors freely available? 

The dataset is not available online. Authors state that the data underlying the results presented in the study are available to authorized researchers and provide a link to apply for authorization to access data.

* Is the manuscript well organized and written clearly enough to be accessible to non-specialists? 

The manuscript is well-written and easy to understand. The English and Scientific language is of adequate quality throughout the manuscript. 

Reviewer #3: Introduction

Please elaborate or define or give examples of internalizing and externalizing problems for the benefit of the reader. Especially since externalizing factor is one of the significant findings in this study.

Methods

It is quite a complicated methodology with various timelines, from participant and mothers as well as various questionnaires used. Perhaps a diagrammatic flowchart will benefit the readers in understanding the process.

A few of the childhood risk factors were elicited at time of child's birth such as birth order and socioeconomic adversity, which may have changed during their lifetime. This step cannot be undone but can be mentioned as a limitation of this study.

Childhood internalizing and externalizing problems: 

1. It is not very clear regarding SACAS. You mentioned in the manuscript:" The South African Child Assessment Schedule (SACAS) was used to ascertain child externalising problems……The SACAS is an 85-item questionnaire based on the Child Behaviour Checklist" and "Externalising problems were assessed with 35 items describing…"

Is the 35 items part of SACAS? 

2. Perhaps can put in the internal reliability (Cronbach alpha) of the SACAS to further strengthen the strength of this tool

Throughout the study, there seems to be different ages at which the assessments took place which does not tally with each other. For example:

- Child externalizing problems were assessed at 5,7, 10 from their mothers.

- Adverse Childhood Experiences (ACEs) were assessed at 5,7, 11

Why is this so?

Adverse Childhood Experience (ACEs)

1. Are the questions to assess ACEs from a reliable and validated questionnaire?

2. "Overall exposure to ACEs was computed by summing the number of reported ACEs (by either mothers or participants) and the final score was standardized" …is not a very clear description of the scoring process.

Reviewer #4: This is a well-written paper that investigates an important public health problem about which little is known, re: childhood risk factors for youth suicide (14-28 years) in a LMIC setting (South Africa). The study data is from a population-based longitudinal prospective study that followed-up children from birth to adulthood. Study methods, measures and data analyses are adequately described. Suicidal ideation (outcome measure) was assessed as self-reported suicidal ideation at any time (at ages 14, 17, 22, and 28 years). Independent measures included early-life adversity (data collected at baseline), children's externalizing and internalising problems (assessed at ages 5, 7, and 10 years, from maternal reports using the SACAS), and ACEs (obtained from mothers at child ages 5, 7, and 11, and from children at ages 11 and 13). The main findings were:

1. 22.3% participants reported SI between ages of 14 and 28; peak SI was at age 17 years

2. Females reported higher SI at all time points except for age 28 when males = females

3. Externalising problems in childhood independently predicted SI

4. Internalising problems in childhood was not an independent predictor of S

5. Childhood socioeconomic adversity was associated with SI in males alone 

6. ACEs, low birth weight, and high birth order, were associated with SI in females alone

The lack of an association between internalizing behaviors and risk of suicide is a major albeit surprising finding that is inconsistent with existing literature (1). This discrepancy raises the question of the extent to which the study (measures, procedures) establishes a trustworthy cause and effect relationship between internalizing behaviors and suicidal ideation. That is, the study's internal validity that makes it possible to eliminate alternative explanations for the finding. Threats to internal validity and thus potential sources of bias include:

1. Use of a parent report measure to assess internalizing behaviors. Parent ratings of children's behaviors may reflect parental attitudes and stress just as much as they reflect the child's behavior

2. Repeated testing of participants using the same measures. This influences study outcomes as participants will often do better as they become used to the testing process 

3. Maturation effects: this refers to the impact of time as a variable in a study (for example, participants growing older). This can make it impossible to rule out whether effects seen in the study were simply due to the effect of time.

4. Attrition: in the study, the analytical sample was a biased one as it differed from the original cohort on many variables, including maternal age and schooling, household crowding, and assets. 

5. Inverse probability weighting: used to address biases due to differential attrition in data analyses. However, one problem issue with IP-weighting is that participants who are extremely unlikely to be treated (that is, those with negative association for SI) will end up with a large weight, potentially making the weighted estimator highly unstable. A common alternative to the conventional weights that at least "kind of" addresses this problem are the stabilized weights, which use the marginal probability of treatment instead of 1 in the weight numerator (2).

I would recommend that the authors address these issues pertaining to internal validity.

References:

1. Auerbach, R. P., Mortier, P., Bruffaerts, R., Alonso, J., Benjet, C., Cuijpers, P., Demyttenaere, K., Ebert, D. D., Green, J. G., Hasking, P., Lee, S., Lochner, C., McLafferty, M., Nock, M. K., Petukhova, M. V., Pinder-Amaker, S., Rosellini, A. J., Sampson, N. A., Vilagut, G., Zaslavsky, A. M., … WHO WMH-ICS Collaborators (2019). Mental disorder comorbidity and suicidal thoughts and behaviors in the World Health Organization World Mental Health Surveys International College Student initiative. International journal of methods in psychiatric research, 28(2), e1752. https://doi.org/10.1002/mpr.1752

2. The intuition behind inverse probability weighting in causal inference https://www.rebeccabarter.com/blog/2017-07-05-ip-weighting/

[LINK]

---

## [Decision Letter · Decision Letter 2]

8 Feb 2022

Dear Dr. Orri,

Thank you very much for re-submitting your manuscript "Childhood factors associated with suicidal ideation among South African youth A 28-year longitudinal study using the Birth to Twenty Plus cohort" (PMEDICINE-D-21-03460R2) for review by PLOS Medicine.

I have discussed the paper with my colleagues and the academic editor and it was also seen again by 3 reviewers. I am pleased to say that provided the remaining editorial and production issues are dealt with we are planning to accept the paper for publication in the journal.

[LINK]

We look forward to receiving the revised manuscript by Feb 15 2022 11:59PM.   

Sincerely,

Caitlin Moyer, Ph.D.

Associate Editor 

PLOS Medicine

plosmedicine.org

Requests from Editors:

1. From the academic editor: The amount of missing data for violence/abuse and for postnatal depression was very high. So, although postnatal depression was not associated with attrition from the cohort, I wonder if it was associated with whether or not data were available on these variables. This may help to explain the lack of association between postnatal depression and later suicidality (although the authors' arguments about why that association may not be seen in this context are also relevant). Please address and comment on the potential relationship between the data that were not available and depression and how this could factor into the absence of a significant association with suicidal ideation.

2. Title: We suggest the following revision to the title. Please make this change in the manuscript submission system as well as the text: “Childhood factors associated with suicidal ideation among South African youth: A 28-year longitudinal study of the Birth to Twenty Plus cohort”

3. Data availability statement: Please update this information in the manuscript submission system (Data availability section) rather than including it at page 20. Please note that there was no non-author contact information provided, as indicated in your response.

4. Response to editor’s comment 20: Please do include Table S1. It is not necessary to include the flowchart, but please do provide the numbers and explanation (e.g. “Of the original 3,273, suicidal ideation data were not available for…” or similar) for those not included in the text of the Methods (approximately at lines 154-155: “For this study, we analyzed data from a sample of 2,020 participants with at least one measure of suicidal ideation at ages 14, 17, 22, or 28 years.”).

5. Abstract: Line 31-32: We suggest revising to: “We documented the associations between individual, familial, and environmental factors in childhood with suicidal ideation…”

6. Abstract: Line 47: Please clarify if p<0.030 should be p=0.03, or otherwise, please report the exact p value unless p<0.001.

7. Author summary: Line 65: Please revise to “To the best of our knowledge, no longitudinal study has been conducted…” or similar.

8. Author summary: Line 72: Please revise to “2,020” in the second bullet point.

9. Author summary: Line 81-83: We suggest revising these two points slightly to avoid suggesting causal implications (e.g. suggesting a direct link to prevention) of the findings.

10. Author summary: Line 84: Please provide slightly more explanation for this point.

11. Introduction: Line 87: Please clarify to “...the second or third most common cause of death…” or similar.

12. Methods: Line 207: Please change the superscript reference to [43] if this is correct.

13. Methods: Line 232: Data analysis plan: Please state that there was no formal prospective analysis plan for the study, but please do mention that the analysis protocol was decided upon during study group meetings (please indicate when this took place, i.e. prior to initiation of the analyses), and please indicate that changes to the planned analyses with rationale (e.g. following peer reviewer comments) are described in the Methods.

14. Results: Line 264: Please change to P=0.007 if this is accurate. Please report the exact p value, unless p<0.001.

15. Results: Line 265 and 266: Please change to “statistically significantly” and “statistically significant” in this sentence. Please also provide the p values for the sex-specific associations reported.

16. Results: Line 279-281: We suggest revising to: “...stratified analyses by sex suggested that the association was stronger among females (OR 1.14, CI 1.05-1.24, P=0.003) than males (OR 1.06, CI 0.95-1.18, P=0.299), although the interaction was not statistically significant…”.

17. Results: Line 288: Please change to P=0.011 if this is accurate. Please report the exact p value, unless P<0.001.

18. Discussion: Line 294: We suggest that the heading “Main Findings” is not necessarily needed.

19. Discussion: Line 299: We suggest revising to: “when rates for males and females were similar…” 

20. Discussion: Lines 402-404: We suggest revising slightly to: “Prevalence rates were higher among females than males, and we found sex differences in the associations of childhood individual, familial, and environmental factors with youth suicidal ideation. As these factors (e.g., externalizing problems, socioeconomic adversity, ACEs) are highly prevalent…”

21. Page 20: Please remove the sections titled Contributions, Conflict of Interest disclosures, and Data Access from the main text. Please make sure that all information is entered completely and accurately into the relevant sections of the manuscript submission system.

22. Table 2: Please define RR in the legend.

23. Table 3 and Table 4: Please define OR in the legend.

24. Figure 1: Please use a color scheme that does not involve differentiating red and green colors.

25. Reference 24: Please change the journal title abbreviation to PLoS One.

26. Reference 51: Please change the journal title abbreviation to Lancet.

27. Reference 53: Please update with complete citation information.

28. Table S1 and S2: Please define SD in the legend.

29. Table S3: Please define ACE in the legend.

Comments from Reviewers:

Reviewer #1: The authors have addressed my concerns and I now recommend publication.

Peter Flom

Reviewer #3: All comments were addressed and relevant amendments made. Thank you.

Reviewer #4: The issues I raised in the earlier review have been satisfactorily addressed by the authors.

[LINK]

---

## [Editor Report · Decision Letter 3]

14 Feb 2022

Dear Dr Orri, 

On behalf of my colleagues and the Academic Editor, Charlotte Hanlon, I am pleased to inform you that we have agreed to publish your manuscript "Childhood factors associated with suicidal ideation among South African youth A 28-year longitudinal study of the Birth to Twenty Plus cohort" (PMEDICINE-D-21-03460R3) in PLOS Medicine.

Please also address the following editorial points:

-Title: Please update the title in the manuscript submission system: “Childhood factors associated with suicidal ideation among South African youth: A 28-year longitudinal study of the Birth to Twenty Plus cohort”

-Methods: Line 157: Please correct to “12,530” if this is accurate.

-Table 1: Please define the abbreviation for ACE in the legend.

PRESS

Sincerely, 

Caitlin Moyer, Ph.D. 

Associate Editor 

PLOS Medicine